# Efflux Pump Inhibitors against Nontuberculous Mycobacteria

**DOI:** 10.3390/ijms21124191

**Published:** 2020-06-12

**Authors:** Laura Rindi

**Affiliations:** Dipartimento di Ricerca Traslazionale e delle Nuove Tecnologie in Medicina e Chirurgia, Università di Pisa, I-56127 Pisa, Italy; laura.rindi@med.unipi.it; Tel.: +39-050-2213-688; Fax: +39-050-2213-682

**Keywords:** efflux pump inhibitor, nontuberculous mycobacteria, drug resistance, *Mycobacterium avium* complex, *Mycobacterium abscessus*

## Abstract

Over the last years, nontuberculous mycobacteria (NTM) have emerged as important human pathogens. Infections caused by NTM are often difficult to treat due to an intrinsic multidrug resistance for the presence of a lipid-rich outer membrane, thus encouraging an urgent need for the development of new drugs for the treatment of mycobacterial infections. Efflux pumps (EPs) are important elements that are involved in drug resistance by preventing intracellular accumulation of antibiotics. A promising strategy to decrease drug resistance is the inhibition of EP activity by EP inhibitors (EPIs), compounds that are able to increase the intracellular concentration of antimicrobials. Recently, attention has been focused on identifying EPIs in mycobacteria that could be used in combination with drugs. The aim of the present review is to provide an overview of the current knowledge on EPs and EPIs in NTM and also, the effect of potential EPIs as well as their combined use with antimycobacterial drugs in various NTM species are described.

## 1. Introduction

Infections by nontuberculous mycobacteria (NTM) represent a relevant problem for human health in many countries in the world. NTM, consisting of all *Mycobacterium* species except for *Mycobacterium tuberculosis* complex and *Mycobacterium leprae*, are a group of over 180 environmental species, generally endowed with low pathogenicity to humans [1,2]. However, over the last years, NTM have emerged as important human pathogens and infections caused by these organisms have increased globally [3,4,5]. Such trend is due to not only more accurate identification tools but also an increased awareness to look for NTM as opportunistic pathogens in clinical specimens; the increased life expectancy and the growing use of immunosuppressive therapies are further factors that may explain the raised incidence rate of NTM infections. NTM are associated with a variety of human diseases, especially concomitantly to particular risk factors; respiratory tract infections are the most frequent, followed by lymphadenitis in children, disseminated infections in severely immunocompromised patients, and skin infections [6,7]. Although significant differences in geographic distribution of NTM have been observed, the most relevant NTM species for human diseases are species belonging to *Mycobacterium avium* complex (MAC) and other slowly growing mycobacteria, such as *Mycobacterium kansasii*, *Mycobacterium xenopi*, and *Mycobacterium malmoense*, and rapidly growing mycobacteria, such as *Mycobacterium abscessus*, *Mycobacterium chelonae*, and *Mycobacterium fortuitum* [8,9,10,11]. In particular, the most frequently reported clinically significant species are MAC and *M. abscessus* [12].

Infections caused by NTM are often difficult to treat since therapy, that needs the use of a combination drug regimen, is long, expensive, and more toxic and more likely to fail than tuberculosis therapy [13,14]. The highly hydrophobic lipid-rich cell wall constitutes a strong barrier to the penetration of various drugs, representing a major contributor for intrinsic resistance of NTM to many antimicrobial compounds currently available [15], thus encouraging an urgent need for the development of new drugs for the treatment of mycobacterial infections [16].

Efflux pumps (EPs) are further important elements that are involved in drug resistance by expelling toxic substrates and thus preventing intracellular accumulation of antibiotics reducing their clinical efficacy [17]. EP activity can also be induced upon drug exposure, thus contributing to both intrinsic and acquired resistance [18]. Several mycobacterial EPs have been identified and characterized to date. The genome of *M. tuberculosis* contains genes encoding drug EPs belonging to all the five superfamilies, such as the ATP-binding cassette (ABC) superfamily, the major facilitator superfamily (MFS), the small multidrug resistance (SMR) superfamily, the resistance-nodulation-cell division (RND) superfamily, and the multidrug and toxic compound extrusion (MATE) superfamily. The ABC superfamily members are primary transporters because they are energized by the hydrolysis of ATP, whereas the other families of EPs are typically energized by the proton motive force and are thus classified as secondary transporters [19]. The presence of several EPs has also been showed in NTM [16]. The best characterized EPs in NTM are those present in *M. fortuitum* conferring resistance to tetracycline and aminoglycoside [20,21], those present in *Mycobacterium smegmatis* conferring resistance to fluoroquinolones, rifamycins, and isoniazid [22,23], those present in MAC conferring macrolide resistance [18,24], and those present in *M. abscessus* conferring resistance to clofazimine and bedaquiline [25]. Overexpression or the increased activity of existing EPs in response to prolonged exposure to the subeffective levels of antimycobacterial drugs, such as may be present during management of the NTM-infected patient, may render an organism increasingly resistant to one or more drugs employed in therapy [26].

A promising strategy to increase drug susceptibility is the inhibition of EP activity by EP inhibitors (EPIs). EPIs are compounds that act on EPs and block their efflux function. EPIs were shown to inhibit efflux of antituberculosis drugs, to decrease *M. tuberculosis* drug resistance and to produce synergistic effects with antimicrobials [27]. Recently, attention has been focused on identifying EPIs in mycobacteria that could be used in combination with drugs [28]. Most studies regarding the use of EPIs to decrease active drug efflux have been performed in *M. tuberculosis*. Some examples of EPIs in *M. tuberculosis* include the following compounds: the Ca^2+^ channel blocker verapamil (VP), which decreased the level of resistance to rifampicin, isoniazid, ofloxacin, streptomycin and to the new anti-TB drugs bedaquiline and clofazimine; phenothiazines, such as thioridazine (TZ) and chlorpromazine (CPZ), which have been proven to reduce clarithromycin and isoniazid resistance in *M. tuberculosis* complex; protonophores, including carbonyl cyanide m-chlorophenylhydrazone (CCCP) and 2,4-dinitrophenol (DNP), which decreased the level of resistance to rifampicin, isoniazid, ofloxacin, and streptomycin; valinomycin, which acts reducing the gradient generated by K^+^ and inhibits novobiocin and rifampin resistance mediated by the P55 pump; plant-derived EPI, such as reserpine (RES), which has been shown to influence the level of resistance to numerous anti-TB drugs, and piperine (PIP), which is found to be synergistic with rifampicin treatment in a mouse infection model of *M. tuberculosis* [29,30].

The aim of the present review is to provide an overview of the current knowledge on EPs and EPIs in NTM and describe the effect of potential EPIs as well as their combined use with antimycobacterial drugs in various NTM species. Table 1 summarizes the best characterized EPs in NTM, whereas Table 2 summarizes compounds investigated as EPIs on NTM in recent years and shows drugs where MIC was decreased by EPI. The following paragraphs will consider the studies carried out on each species of NTM.

## 2. *Mycobacterium avium* Complex (MAC)

Among the NTM, the members of the MAC are responsible for most of the human-associated infections and can cause chronic pulmonary infections in adults, lymphadenitis in children, and disseminated infections in immunocompromised subjects [53]. MAC is a group of slow-growing mycobacteria, consisting of 12 species: *Mycobacterium avium*, consisting of 4 subspecies, i.e., *M. avium* subsp. *hominissuis*, *M. avium* subsp. *avium*, *M. avium* subsp. *silvaticum*, and *M. avium* subsp. *paratuberculosis*, *Mycobacterium intracellulare*, *Mycobacterium chimaera*, *Mycobacterium colombiense*, *Mycobacterium arosiense*, *Mycobacterium vulneris*, *Mycobacterium bouchedurhonense*, *Mycobacterium timonense*, *Mycobacterium marseillense*, *Mycobacterium yongonense*, *Mycobacterium paraintracellulare*, and *Mycobacterium lepraemurium* [54]. The MAC species most frequently isolated from human infections are *M. avium*, *M. intracellulare*, and *M. chimaera*. *M. avium* is frequently responsible for disseminated disease in patients with acquired immunodeficiency syndrome (AIDS), whereas *M. intracellulare* appears more likely to cause pulmonary infections in immunocompetent patients [55]. Macrolides, such as clarithromycin, are key drugs for treatment of MAC infections. The development of clarithromycin resistance, that makes the therapy prone to fail [56], has been mainly attributed to mutational alterations of the 23S rRNA gene and to overexpression of EPs [15]. In particular, the upregulation of EP activity, induced upon drug exposure, can significantly decrease the intracellular concentration of clarithromycin, reducing its clinical efficacy.

Genomic analysis of *M. avium* showed the presence of several genes coding for EPs belonging to almost all the known superfamilies such as ABC, MFS, RND, and SMR. Notably, members of MAC presented a lot of the mycobacterial membrane protein Large (MmpL) proteins, a subclass of RND transporters known to participate in the export of lipid components across the cell envelope; *M. avium*, *M. intracellulare*, and *M. chimaera* have 14, 18, and 21 MmpL, respectively [57]. Mutations in the *mmpT5* encoding TetR repressor located ahead of the *mmpS5/mmpL5* operon in *M. intracellulare* were associated with modest resistance levels to bedaquiline and clofazimine [31]. It was demonstrated that there are specific genes that encode EPs associated with macrolide resistance in *M. avium*, i.e., *MAV_3306*, encoding an ABC transporter and *MAV_1406*, encoding a MFS transporter, which are highly conserved in other pathogenic mycobacteria, are induced in a stepwise manner after subtherapeutic azithromycin exposure [18]. Moreover, the contribution of the *M. avium* EPs MAV_1406 and MAV_1695, belonging to the MFS and ABC transporters, respectively, was proven because they were found to be overexpressed in two isogenic strains showing high-level clarithromycin resistance [24].

The first EPIs investigated for their activity on *M. avium* were phenothiazines, inhibitors of K^+^ transport and Ca^2+^ channels [58]. A study of in vitro activity of CPZ, TZ, promazine, promethazine, and desipramine against a reference and clinical strains of *M. avium* showed TZ as the most active of the group [37]. However, only later, the role of some phenothiazines in the inhibition of EPs was confirmed. Rodrigues et al. [41,42] demonstrated that intrinsic efflux activity of MAC strains was inhibited by TZ and CPZ as well as by VP, a Ca^2+^ channel blocker, and CCCP, a protonophore reducing the transmembrane potential. Accumulation of the known EPs substrate ethidium bromide (EtBr) was found to be significantly increased by the above EPIs in a concentration-dependent manner. At half of their intrinsic MIC, both TZ and CPZ, similarly to VP and CCCP, significantly increased the susceptibility of a reference strain of *M. avium* to erythromycin, suggesting an effect on an EP with EtBr and erythromycin as substrates; moreover, since the susceptibility to amikacin and ethambutol was increased by TZ and CCCP, respectively, and not by CPZ and VP, resistance to these antibiotics may be due to EPs other than the one that extrudes erythromycin and which is affected by all the EPIs tested [41]. A following study on *M. avium* and *M. intracellulare* clinical strains from AIDS patients showed that resistance to clarithromycin was significantly reduced in the presence of TZ, CPZ, and VP, and the same EPIs were effective in decreasing the efflux of EtBr from mycobacterial cells; moreover, increased retention of [^14^C]-erythromycin in the presence of these EPIs further demonstrated that active efflux contributes to MAC resistance to macrolides [42]. A further study evaluated TZ as chemotherapy for MAC diseases by investigating its pharmacokinetics/pharmacodynamics against a reference strain of *M. avium* [59].

In 2015, Machado et al. [24] studied a group of 2-phenylquinoline-derived compounds previously reported as potent inhibitors of the EP NorA of *Staphylococcus aureus* [60]. All four compounds tested were able to inhibit EPs in *M. avium*; in particular, *O*-ethylpiperazinyl derivative 2 showed an efflux inhibitory activity comparable to that of VP and was able to significantly reduce the MIC values of macrolides against wild-type as well as resistant *M. avium* strains. Moreover, the above compound showed synergistic activity with clarithromycin against *M. avium*-infected macrophages at a concentration below the cytotoxicity threshold. Finally, by using *M. avium* isogenic strains presenting high level clarithromycin resistance, a correlation between the synergistic effect of the 2-phenylquinoline with macrolides and the inhibition of EP, most likely the MFS pump MAV_1406, was demonstrated [24]. Subsequently, a series of 3-phenylquinolone, designed by modifying the natural isoflavone biochanin A, was showed to be very effective EPIs of different wild-type and resistant strains of *M. avium*, bringing down the MIC values of macrolides and fluoroquinolones, thus representing the first specifically designed compounds exhibiting potent NTM EPI activity. Interestingly, the 3-phenylquinolones 1e and 1g boosted the activity of antimycobacterial drugs and EtBr against all the *M. avium* strains, decreasing the MIC values up to very low concentrations, showing a better activity than EPIs VP, CPZ, and TZ [43]. Further efforts were made to improve these compounds, in order to overcome their toxicity toward human macrophages. New 3-phenylquinolone analogues were designed, synthesized, and evaluated. Most compounds were active as *M. avium* EPIs, with some of them synergizing with clarithromycin, ciprofloxacin, and EtBr; among these, compounds 11b, 12b, and 16a were active at concentration below their CC_50_ on human cells. Interestingly, derivative 16a was the most promising compound, showing a significant synergistic activity with clarithromycin at a concentration that was sixfold lower than its CC50, thereby representing the best MAC EPI identified. Moreover, this promising compound showed good activity in *M. avium*-infected macrophages both alone as well as in combination with clarithromycin, suggesting its high therapeutic potential [44].

Recently, the effect of EPIs further on clarithromycin susceptibility levels in 12 clinical isolates of *M. avium* and *M. intracellulare* was investigated. It was shown that berberine (BER), CCCP, PIP, and tetrandrine (TTR) were active against both clarithromycin-susceptible and -resistant strains, reducing the MIC of clarithromycin; in particular, the MIC of clarithromycin showed at least a fourfold reduction in presence of BER (83% of total isolates), CCCP (67%), PIP (25%), and TTR (75%). Notably, among the six resistant isolates, TTR reversed the resistance phenotype of three strains, BER of two strains, and CCCP and PIP of one strain [45].

## 3. *Mycobacterium abscessus*

*M. abscessus* is a rapidly growing mycobacterium associated with several diseases in humans, of which lung disease is the most common, particularly in patients with underlying lung disease such as bronchiectasis, chronic obstructive pulmonary disease, and cystic fibrosis; it is the second most common cause of NTM pulmonary disease in many countries after MAC [4]. *M. abscessus* consists of three subspecies, namely, *M. abscessus* subsp. *abscessus*, *M. abscessus* subsp. *bolletii*, and *M. abscessus* subsp. *massiliense* [61]. The poor treatment outcomes of *M. abscessus* diseases are attributable to the resistance to clarithromycin, including acquired resistance, mainly related to mutations in the *rrl* gene, and inducible resistance conferred by the *erm* (41) gene, present in *M. abscessus* subsp. *abscessus* and subsp. *bolletii* but absent in *M. abscessus* subsp. *massiliense* [62]. Therapeutic regimens are based not only on a macrolide but also on an aminoglycoside and a β-lactam [53]; recently, the introduction of clofazimine and bedaquiline is gaining increasing interest for the treatment of *M. abscessus* pulmonary disease.

A high number of MmpL transporters are found in *M. abscessus* [57]. Resistance to thiacetazone derivatives with potent activity against *M. abscessus* was found in spontaneous resistant mutants where genes encoding one of three MmpS5/MmpL5 EP systems, i.e., *MAB_4383c*-*MAB_4282c*, were upregulated [32]. The presence of two other putative MmpS5/MmpL5-coding operons in *M. abscessus* may represent a resistance mechanism through efflux of further antibiotics with anti-*M. abscessus* activity. Moreover, the presence of multiple efflux pumps of the MmpL family that participate to low-resistance levels to clofazimine and bedaquiline was reported [25]. In addition, an association between overexpression of the MFS pumps MAB_3142 (homologue of *M. tuberculosis* P55) and MAB_1409 (homologue of *M. avium* MAV_1406) and clarithromycin resistance was demonstrated in all the three subspecies of *M. abscessus* [33]. Finally, in a recent study, three putative EPs, i.e., an ATP binding cassette EP encoded by *MAB_2355c* (homologue of *M. avium* MAV_3306), a tap like EP encoded by *MAB_1409c*, and an ATP binding cassette EP encoded by *MAB_1846*, as well as their positive regulatory gene *whiB7*, contributed to clarithromycin resistance exhibited by *M. abscessus* [34].

It has been reported that usnic acid, a natural compound found in lichens which has shown antimicrobial activity against *M. tuberculosis* and NTM [63], increased activity of antimicrobials in *M. abscessus*; in particular, usnic acid was able to reduce the MIC of amikacin and clarithromycin fourfold and was effective on EtBr accumulation, thus suggesting its efflux-inhibiting activity [46]. Based on the therapeutic and pharmacological properties of flow regulators of calcium ions of tetrahydropyridine (THP), recent studies evaluated THP compounds as EPIs in *M. abscessus*, showing that some of such compounds were able to reduce the MIC of amikacin, ciprofloxacin, and clarithromycin in a reference strain and in clinical isolates of *M. abscessus* subsp. *abscessus* and subsp. *bolletii*; therefore, THP derivatives may represent potential adjuvants in the treatment of infections caused by *M. abscessus* [47,48]. Interestingly, computational simulations revealed possible binding sites for the derivative NUUM01 on MmpL5 and Tap protein structure [47]. The same authors strengthened the hypothesis that efflux activity plays a role in *M. abscessus* resistance to clarithromycin by demonstrating an increase of mRNA expression levels of the EP MAB_3142 and MAB_1409 in *M. abscessus* strains after exposure to clarithromycin; moreover, they showed that VP increased the susceptibility to clarithromycin from fourfold to >64-fold in nine clinical strains of *M. abscessus* belonging to the T28 *erm* (41) sequevar responsible for the inducible resistance to clarithromycin [33]. More recently, consistent with that reported by Vianna et al. [33], three EPs, including MAB_1409, associated with the intrinsic resistance of *M. abscessus* to clarithromycin were identified; the addition of EPIs phenylalanine-arginine β-naphthylamide (PAβN), which is a peptidomimetic compound, CCCP, and VP significantly decreased the MIC of clarithromycin for resistant *M. abscessus* clinical strains overexpressing MAB_1409 [34].

## 4. *Mycobacterium smegmatis*

*M. smegmatis* is a rapidly growing nonpathogenic mycobacterium, often used as a model organism that offers safety and reduced time for carrying out experiments. The *M. smegmatis* genome contains many genes encoding putative drug EPs that are expressed at detectable levels [22]. Several studies have been conducted on *M. smegmatis* with the aim of preliminary investigation on EPs and inhibitors to be subsequently evaluated on NTM or *M. tuberculosis* [64].

A well-characterized *M. smegmatis* EP is the MFS transporter LfrA, a multidrug EP that contributes to low-level resistance to fluoroquinolones and other toxic compounds such as ethidium bromide [35,36,37]. Li et al. [22] strengthened these features by demonstrating the involvement of LfrA in the intrinsic drug resistance, and also identified further putative EPs. Indeed, it has been proven that the deletions of the *lfrA* gene or *mmr* orthologue rendered the mutant more susceptible to multiple drugs such as fluoroquinolones, ethidium bromide, and acriflavine; the deletion of the *efpA* orthologue also produced increased susceptibility to these agents but also resulted in decreased susceptibility to rifamycins, isoniazid, and chloramphenicol; deletion of the *Rv1877* orthologue produced some increased susceptibility to ethidium bromide, acriflavine, and erythromycin [22]. Moreover, De Rossi et al. [38] characterized Tet(V), a tetracycline efflux protein, belonging to the MFS of efflux proteins, and the corresponding gene in *M. smegmatis*; preliminary evidence suggests that a homologous gene is present in *M. fortuitum* but not in other mycobacteria [38]. More recently, the MSMEG_2631 gene (*mmp*) encoding a MATE family protein in *M. smegmatis* was characterized, and data reported that *mmp* deletion increased susceptibility for phleomycin, bleomycin, capreomycin, amikacin, kanamycin, cetylpyridinium chloride, and several sulfa drugs [39]. In another study, overexpression of the two ABC transporter genes *Ms6509* and *Ms6510* remarkably decreased mycobacterial rifampicin resistance and deletion of these two transporter genes resulted in increased rifampicin resistance in *M. smegmatis* [40]. Finally, *M. smegmatis* presents a high number of MmpL transporters [57].

In the strain *M. smegmatis* mc^2^155, it was demonstrated that the accumulation of EtBr was found to be significantly increased by EPIs TZ, CPZ, VP, and CCCP in a concentration-dependent manner; moreover, VP considerably increased the susceptibility of *M. smegmatis* to erythromycin [41]. Subsequently, Rodrigues et al. [52], in a study aimed to describe EtBr transport in *M. smegmatis* using the wild-type strain mc^2^155 and mutant strains carrying deletions of genes coding for porins and the LfrA pump, reported an overall reduction of MIC of streptomycin, rifampicin, amikacin, ciprofloxacin, clarithromycin, and erythromycin caused by CPZ, TZ, and VP in most of the strains tested. The fact that the effect of these EPIs was not dependent of a given genotype suggested that these inhibitors have a wide range of activity against efflux and are not specific for a particular EP [52]; moreover, the mutant for the LfrA pump showed increased accumulation of EtBr and increased susceptibility to EtBr, ethambutol, and ciprofloxacin, in agreement with previous studies [35,36]. In 2015, Machado et al. [24] studied a group of 2-phenylquinoline-derived compounds previously reported as inhibitors of the EP norA of *Staphylococcus aureus* [60]. All four compounds tested were able to promote accumulation of EtBr in *M. smegmatis* mc^2^155, and *O*-ethylpiperazinyl derivative 2 showed a highly synergistic activity with clarithromycin [24].

In order to identify new antimycobacterial natural compounds from plants, Lechner et al. screened flavonoids as EPIs in *M. smegmatis* [49,65]. Each tested plant natural product, namely, epicatechin, isorhamnetin, kaempferol, luteolin, myceritin, quercitin, rutin, and taxifolin, caused at least a twofold decrease of the MIC of isoniazid at subinhibitory concentrations in *M. smegmatis* reference strains; among such compounds, myceritin was the most efficient intensifier of isoniazid susceptibility causing a reduction of the MIC of isoniazid up to 64-fold, followed by quercitin; myceritin did not decrease the MIC values of other antibiotics, suggesting the synergistic activity and specificity for isoniazid potentiation [49]. Moreover, the isoflavone biochanin A has been demonstrated to exhibit EP-inhibiting activity in *M. smegmatis* mc^2^155 comparable to that of reference EPI controls, such as VP, by performing an EtBr efflux assay [65]; this last study prompted further investigations of this class of compounds as efflux inhibitors against *M. smegmatis* and *M. avium*, as previously described [43,44]. In particular, a series of 3-phenylquinolone, designed by modifying the natural isoflavone biochanin A, showed moderate to good synergistic effects against *M. smegmatis* when combined with clarithromycin and ciprofloxacin [44]. Additional natural plant metabolites were studied for their EPI activity against *M. smegmatis*. Farnesol, an isoprenoid component, significantly enhanced the accumulation of EtBr, blocking its efflux, thus suggesting that farnesol act as an EPI in *M. smegmatis*. Interestingly, farnesol also showed synergism when combined with rifampicin [51]. The plant alkaloid PIP extensively increased accumulation and decreased the efflux of EtBr in *M. smegmatis*, which suggest that it has an ability to inhibit mycobacterial EPs. PIP achieved EP inhibition levels comparable to the RES standard EPI control, and the EtBr efflux inhibition ability of PIP was greater than CPZ but lower than CCCP; moreover, higher concentration of PIP exhibited a stronger EtBr efflux inhibitory effect, demonstrating that PIP inhibited efflux in a concentration-dependent manner [66]. A very recent study evaluated the antimicrobial and resistance-modifying profile of a range of plant-derived flavonoids, such as skullcapflavone, nobiletin, tangeretin, baicalein, and wogonin; after a screening of all compounds for their synergistic effects with EtBr and rifampicin, authors concluded that skullcapflavone II considerably increased the susceptibility of *M. smegmatis* to EtBr and rifampicin and that nobiletin was found to be the most potent efflux inhibitor [50].

## 5. Other NTM Species

A well-characterized EP in NTM is the Tap, an EP of the MFS superfamily, present in *M. fortuitum* conferring resistance to tetracycline and aminoglycoside [20]. Tap protein from *M. fortuitum* uses the electrochemical gradient across the cytoplasmic membrane to extrude tetracycline from the cell; this efflux activity is inhibited by CCCP and RES, consistent with the decrease in MIC observed in antibiotic susceptibility testing in the presence of these inhibitors. Moreover, CCCP, RES, and CPZ reduced the MIC of tetracycline in a *M. smegmatis* strain expressing the Tap protein of *M. fortuitum* [21]. A study performed to detect LfrA and Tap EPs among 166 clinical isolates of rapidly growing mycobacteria demonstrated that *lfrA* is rare, whereas *tap* is found more commonly [67]. In fact, the *lfrA* gene was detected in 4 strains (2.4%), comprising 2 strains of *Mycobacterium chelonae*, 1 *Mycobacterium fortuitum*, and 1 *Mycobacterium mucogenicum*; on the other hand, the *tap* gene was detected in 109 strains (65.7%), comprising 3 *Mycobacterium abscessus*, 12 *M. chelonae*, 75 *M. fortuitum*, 2 *Mycobacterium mageritense*, 15 *Mycobacterium peregrinum*, 1 *Mycobacterium alvei*, and 1 *Mycobacterium porcinum*. No correlation was detected between the presence of the EPs and resistance to quinolones or tetracyclines, suggesting that other mechanisms must be involved in antibiotic resistance of these strains [67]. A recent review, comparing the MmpL abundance among different mycobacterial species, reported a higher number of MmpLs in rapid growing mycobacteria when compared with slow-growing mycobacteria. Notably, among the rapid growing mycobacteria, *M. fortuitum*, together with *M. abscessus* and *M. smegmatis*, has the highest number of MmpLs [57].

Plant-derived flavonoids, such as epicatechin, isorhamnetin, kaempferol, luteolin, myceritin, quercitin, rutin, and taxifolin, were tested for their ability to decrease the MIC of isoniazid, not only on *M. smegmatis* but also on references strains of *M. phlei* and *M. fortuitum*. Myceritin was the most efficient EPI, decreasing the MIC of isoniazid 8-fold in *M. fortuitum* and 16-fold in *M. phlei* [49]. Moreover, a following study reported the investigation on further flavonoids, namely, skullcapflavone, nobiletin, tangeretin, baicalein, and wogonin in a reference strain of *Mycobacterium aurum*. Nobiletin was shown to be the most effective promoter for EtBr accumulation, with a similar effect to that of VP; skullcapflavone and tangeretin enhanced EtBr-accumulation to a greater extent than CPZ but were less effective than VP; baicalein and wogonin were the least effective of the compounds tested [50].

## 6. Conclusions

Recent researches have shown that EPIs may represent promising adjuvants to conventional antimycobacterial therapy by enhancing susceptibility to drugs. Several compounds, both known EPIs and new natural plant-derived molecules, have been proven to be very active in increasing the efficacy of many drugs in NTM, even in resistant strains. However, the implementation of EPIs in clinical practice has been hampered by the toxic properties and the pharmacological effects of these inhibitors in human host. Further advanced investigations on this topic are needed since the development of combinational therapies using EPIs and antibiotics could constitute a novel approach that should be considered to control NTM infections.

## Figures and Tables

**Table 1 ijms-21-04191-t001:** Major nontuberculous mycobacteria (NTM) efflux pumps (EPs) associated with drug resistance.

NTM Species	EP Gene	EP Superfamily ^a^	Drugs ^b^	References
**MAC**	*MAV_3306*	ABC	AZT	[18]
	*MAV_1406*	MFS	AZT, CLA	[18,24]
	*MAV_1695*	ABC	CLA	[24]
	*MAV_2510 (mmpL5)*	RND	BDQ, CLO	[31]
***M. abscessus***	*MAB_2301* (*mmpL*)	RND	BDQ, CLO	[25]
	*MAB_1134c* (*mmpL*)	RND	BDQ, CLO	[25]
	*MAB_4382c (mmpl5)*	RND	TAC derivatives	[32]
	*MAB_3142*	MFS	CLA	[33]
	*MAB_1409*	MFS	CLA	[33,34]
	*MAB_2355*	ABC	CLA	[34]
	*MAB_1846*	ABC	CLA	[34]
***M. smegmatis***	*lfrA*	MFS	Fluoroquinolones, TET	[22,35,36,37]
	*emrE*	SMR	Fluoroquinolones	[22]
	*efpA*	MFS	Fluoroquinolones, rifamycins, INH, CHL	[22]
	*Rv1877* orthologue	MFS	ERY, TET, KAN	[22]
	*TetV*	MFS	TET	[38]
	*MSMEG_2631*	MATE	CAP, AMI, KAN	[39]
	*Ms6509*	ABC	RIF	[40]
	*Ms6510*	ABC	RIF	[40]
***M. fortuitum***	*tap*	MFS	TET, aminoglycoside	[20,21]

^a^ EP superfamily: ABC, ATP-binding cassette superfamily; MFS, major facilitator superfamily; RND, resistance-nodulation-cell division superfamily; SMR, small multidrug resistance superfamily; MATE, multidrug and toxic compound extrusion superfamily. ^b^ Drugs: AZT, azithromycin; CLA, clarithromycin; BDQ, bedaquiline; CLO, clofazimine; TAC, thiacetazone; TET, tetracycline; INH, isoniazid; CHL, chloramphenicol; ERY, erythromycin; KAN, kanamycin; CAP, capreomycin AMI, amikacin; RIF, rifampicin.

**Table 2 ijms-21-04191-t002:** Combinations of efflux pumps inhibitors (EPIs) and drugs evaluated in NTM species, where EPIs decreased the minimal inhibitory concentrations (MICs) of the drugs.

NTM Species	Combination EPI/Drugs ^a^	References
MAC		
*M. avium* *ATCC* 25291 104	CPZ/ERY, CLA	[41,42]
	TZ/ERY, AMI, CLA	[41,42]
	VP/ERY, CLA	[41,42]
	CCCP/ERY, EMB	[41]
	2-phenylquinolines/CLA, ERY, AZT	[24]
	3-phenylquinolones/CLA, ERY, AZT, CPX, OFX	[43,44]
*M. intracellulare* *ATCC* 13950	CPZ/CLA, ERY	[42]
	TZ/CLA, ERY	[42]
	VP/CLA, ERY	[42]
*M. avium* and *M. intracellulare* *clinical* strains	CPZ/CLA, ERY	[42]
	TZ/CLA, ERY	[42]
	VP/CLA, ERY	[42]
	2-phenylquinolines/CLA, ERY, AZT	[24]
	3-phenylquinolones/CLA, ERY, AZT, CPX, OFX	[43]
	BER/CLA	[45]
	CCCP/CLA	[45]
	PIP/CLA	[45]
	TTR/CLA	[45]
***M. abscessus***		
*M. abscessus* subsp. *abscessus* ATCC19977	Usnic acid/AMI, CLA	[46]
	THP compounds/AMI, CLA, CPX	[47,48]
	VP/CLA	[33]
*M. abscessus* subsp. *abscessus* clinical strains	Usnic acid/AMI, CLA	[46]
	THP compounds/AMI, CLA, CPX	[47,48]
	VP/CLA	[33,34]
	CCCP/CLA	[34]
	PaβN/CLA	[34]
*M. abscessus* subsp. *bolletii* clinical strains	Usnic acid/AMI, CLA	[46]
	THP compounds/AMI, CLA, CPX	[48]
	VP/CLA	[33]
*M. abscessus* subsp. *massiliense* clinical strains	VP/CLA	[33,34]
	CCCP/CLA	[34]
	PaβN/CLA	[34]
***M. smegmatis***		
*M. smegmatis* mc^2^155, mc^2^2700, ATCC 14468	Flavonoids/INH, RIF	[49,50]
	Farnesol/RIF	[51]
	CPZ/RIF, AMI, STR, CPX, CLA, ERY	[52]
	TZ/RIF, AMI, STR, CPX, CLA, ERY	[52]
	VP/RIF, AMI, STR, CPX, CLA, ERY	[41,52]
	2-phenylquinolines/CLA	[24]
	3-phenylquinolones/CLA, CPX	[44]
***M. aurum***		
*M. aurum* ATCC 23366	Flavonoids/RIF	[50]
***M. fortuitum***		
*M. fortuitum* ATCC 6841	Flavonoids/INH	[49]
***M. phlei***		
*M. phlei* ATCC 11758	Flavonoids/INH	[49]

^a^ EPI: CPZ, chlorpromazine; TZ, thioridazine; VP, verapamil; CCCP, carbonyl cyanide m-chlorophenylhydrazone; BER, berberine; PIP, piperine; TTR, tetrandrine; THP, tetrahydropyridine; PaβN, phenylalanine-arginine β-naphthylamide. Drugs: ERY, erythromycin; CLA, clarithromycin; AMI, amikacin; EMB, ethambutol; AZT, azithromycin; CPX, ciprofloxacin; OFX, ofloxacin; INH, isoniazid; RIF, rifampicin.

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
