# Peer review of "Efflux Pump Inhibitors against Nontuberculous Mycobacteria"

_ijms, 2020, doi:10.3390/ijms21124191_

Round 1

Reviewer 1 Report

General Comments

  1. A brief review of the possible role of efflux pump inhibitors for combination antibiotic therapy of NTM
  2. Sadly, this is another review in a crowded field; every microbiological journal has a like review 
  3. Tabular presentation of the current data is a strong point

Specific Comments

Abstract

Lines 9-10. I don't think this sentence lacks background, namely the role of the lipid-rich outer membrane in multidrug resistance, and could lead a reader to think of plasmid-encoded multidrug resistance.

Introduction

Line 25. "ubiquitous" is unnecessary and a bit misleading as some species are geographically constrained.

Lines 41-47. Please consider reversing the two sentences, as it is important to have a thorough understanding of "multidrug-resistance" in mycobacteria. They are innately resistant because of the presence of the impermeable lipid-rich outer membrane.

Line 47. Please start a new paragraph.

Line 66. What is meant by "reverse acquired resistance"? Does the phrase refer to spontaneous mutation or innate ability to adapt by gene expression to novel conditions?

Lines 67-68. Please cite the evidence that efflux pump inhibitors "are able to increase the intracellular concentration of antimicrobials". The information that follows simply lists combinations of inhibitors and drugs that have led to increased susceptibility.

Table 1, Lines 88-94. In Table 1, the second heading is "EP", shouldn't that be EP Gene?

Table 2, Lines 95-96. The title of Table 2 is incorrect as there is no listing of "Effects" of efflux pump inhibitors, simply combinations. Please re-label the columns as they are vague.

Table 2, Lines 95-96. All entries under "Drugs" require references

Lines 101-113, Mycobacterium avium complex (MAC). The taxonomy of this group is inadequately covered and presented. For example, there is no listing of the subspecies, nor of the other 8 species. There is no mention of Mycobacterium chimaera, a close relative of M. intracellulare and an important pathogen in its own right.

Line 148. If norA is a gene, it should be italicized.

Lines 226-251, Mycobacterium smegmatis. Sadly, M. smegmatis is a lab rat and results in that Mycobacterium do not provide any guidance for other pathogenic mycobacteria. Its weaknesses outweigh its strength of rapid growth and Biosafety Level 1 biocontrol. Its constraints should be underscored as too many drugs with activity in M. smegmatis have failed in other mycobacteria.

Line 252. M. smegmatis mc2a55 is not a type strain. It is a spontaneous transformable mutant of the type strain M. smegmatis strain 607.

Author Response

Responses to Reviewer 1:

  1. Lines 9-10. I don't think this sentence lacks background, namely the role of the lipid-rich outer membrane in multidrug resistance, and could lead a reader to think of plasmid-encoded multidrug resistance.

This has been corrected in lines 9-10: “an intrinsic multidrug resistance for the presence of a lipid-rich outer membrane”

  1. Line 25. "ubiquitous" is unnecessary and a bit misleading as some species are geographically constrained.

“Ubiquitous” has been deleted, as requested (line 25)

  1. Lines 41-47. Please consider reversing the two sentences, as it is important to have a thorough understanding of "multidrug-resistance" in mycobacteria. They are innately resistant because of the presence of the impermeable lipid-rich outer membrane.

The sentences have been reversed in lines 43-46 and two references have been changed: “The highly-hydrophobic lipid-rich cell wall constitutes a strong barrier to the penetration of various drugs, representing a major contributor for intrinsic resistance of NTM to many antimicrobial compounds currently available [15], thus encouraging an urgent need for the development of new drugs for the treatment of mycobacterial infections [16].”

  1. Line 47. Please start a new paragraph.

A new paragraph has been started in line 47.

  1. Line 66. What is meant by "reverse acquired resistance"? Does the phrase refer to spontaneous mutation or innate ability to adapt by gene expression to novel conditions?

In order to avoid misunderstandings, I changed the statement in line 66: “to increase drug susceptibility”

  1. Lines 67-68. Please cite the evidence that efflux pump inhibitors "are able to increase the intracellular concentration of antimicrobials". The information that follows simply lists combinations of inhibitors and drugs that have led to increased susceptibility.

I rewrote the sentence and included a reference on lines 67-69: “EPIs are compounds that act on EPs and block their efflux function. EPIs were shown to inhibit efflux of anti-tuberculosis drugs, to decrease M. tuberculosis drug resistance and to produce synergistic effects with antimicrobials [27].”

  1. Table 1, Lines 88-94. In Table 1, the second heading is "EP", shouldn't that be EP Gene?

As suggested, in Table 1 the second heading has been changed in “EP gene” and all the names of genes have been italicized.

  1. Table 2, Lines 95-96. The title of Table 2 is incorrect as there is no listing of "Effects" of efflux pump inhibitors, simply combinations. Please re-label the columns as they are vague.

As suggested, the title of Table 2 has been changed as follows: “Combinations of EPIs and drugs evaluated in NTM species, where EPIs decreased the MICs of the drugs”. I also re-labelled and modified the columns.

  1. Table 2, Lines 95-96. All entries under "Drugs" require references

References have been added in Table 2.

  1. Lines 101-113, Mycobacterium avium complex (MAC). The taxonomy of this group is inadequately covered and presented. For example, there is no listing of the subspecies, nor of the other 8 species. There is no mention of Mycobacterium chimaera, a close relative of M. intracellulare and an important pathogen in its own right.

The requested information has been provided (lines 107-113): “MAC is a group of slow-growing mycobacteria, consisting of 12 species: Mycobacterium avium, consinsting of 4 subspecies, i.e., M. avium subsp. hominissuis, M. avium subsp. avium, M. avium subsp. silvaticum and M. avium subsp. paratuberculosis, Mycobacterium intracellulare, Mycobacterium chimaera, Mycobacterium colombiense, Mycobacterium arosiense, Mycobacterium vulneris, Mycobacterium bouchedurhonense, Mycobacterium timonense, Mycobacterium marseillense, Mycobacterium yongonense, Mycobacterium paraintracellulare and Mycobacterium lepraemurium [32]. The MAC species most frequently isolated from human infections are M. avium, M. intracellulare and M. chimaera.”

  1. Line 148. If norA is a gene, it should be italicized.

NorA is a protein, therefore it was not written in italics, but with the first capital letter (line 156)

  1. Lines 226-251, Mycobacterium smegmatis. Sadly, M. smegmatis is a lab rat and results in that Mycobacterium do not provide any guidance for other pathogenic mycobacteria. Its weaknesses outweigh its strength of rapid growth and Biosafety Level 1 biocontrol. Its constraints should be underscored as too many drugs with activity in M. smegmatis have failed in other mycobacteria.

I am aware, as pointed out by the Reviewer, that M. smegmatis is a non-pathogenic mycobacterium susceptible to drugs ineffective against other mycobacteria. However, since M. smegmatis has been used extensively as a model system for M. tuberculosis and other pathogenic mycobacteria, I consider it appropriate to include in the review the description of the investigations and findings obtained with M. smegmatis.

  1. Line 252. M. smegmatis mc2a55 is not a type strain. It is a spontaneous transformable mutant of the type strain M. smegmatis strain 607.

The word “reference” referring to the M. smegmatis strain has been deleted (line 260).

Reviewer 2 Report

In this article entitled " Efflux Pump Inhibitors Against Nontuberculous Mycobacteria", the author did a thorough review of the current state of knowledge on the activity of EPI against NTM. They included in this review MAC, M. abscessus, M. smegmatis and other clinically relevant NTM. I do believe this review is relevant to the field and should be published after minor revisions. I will also suggest the author to revise the manuscript for spellchecks and abbreviations (they were not consistently used throughout the article).    Please find below my comments.

Major comments :

  • Table 2: This Table is hard to read as it is hard to know which EPI, Drugs and REF are linked with which NTm species. I will highly recommend the author to modify this Table. Also, the section on M. smegmatis is hard to read and understand. I will also suggest the author to rewrite this section.
  • MAC section: The author should specify in the text that M. avium they are discussing is M. avium subsp. avium. I think there is confusion coming from the fact that MAH is discussed in lines 106-108. The author should revised this section.

Minor comments :

  • Table 1: The author should give the gene number for MmpL5 in MAC and M. abscessus sections. For M. smegatis section, please change « Rv1877 homologue » for « Rv1877 orthologue ».
  • Line 93: please change chlormphenical to chloramphenicol and kanamicyn to kanamycin.
  • Line 97: By convention, the abbreviation TET is for tetracyclin. Can the author change this abbreviation?
  • Line 115-118: Did the number of MmpL refer in the manuscript are for MmpL only or MmpL and MmpS?
  • Line 127-128: pleas add a reference
  • Lines 192-198: The author should add the MAB gene number of the efflux pump they are refereeing to. As M. abscessus contained several orthologues of the MmpL5/Mmp5 efflux, this will help the reader to understand these sentences.
  • Lines 236-241: please change homologue for orthologue
  • Line 302-303: please add a reference

Author Response

Responses to Reviewer 2:

  1. Table 2: This Table is hard to read as it is hard to know which EPI, Drugs and REF are linked with which NTM species. I will highly recommend the author to modify this Table. Also, the section on M. smegmatis is hard to read and understand. I will also suggest the author to rewrite this section.

Table 2 has been modified. A single column to indicate the EPI/drug combination has been put and the references have been detailed. Moreover, the lines indicating each NTM species have been separated by bold lines. The M. smegmatis section has been rewritten and standardized to the other sections.

  1. MAC section: The author should specify in the text that M. avium they are discussing is M. avium subsp. avium. I think there is confusion coming from the fact that MAH is discussed in lines 106-108. The author should revised this section.

In the revised manuscript the M. avium subspecies is never specified, since many articles cited do not take it into consideration. The sentence on MAH has been removed to avoid confusion (line 113).

  1. Table 1: The author should give the gene number for MmpL5 in MAC and M. abscessus sections. For M. smegatis section, please change « Rv1877 homologue » for « Rv1877 orthologue ».

As suggested, the gene number for Mmpl5 has been provided and “homologue” has been changed in “orthologue”

  1. Line 93: please change chlormphenical to chloramphenicol and kanamicyn to kanamycin.

They have been changed as suggested (line 94).

  1. Line 97: By convention, the abbreviation TET is for tetracyclin. Can the author change this abbreviation?

The abbreviation for tetrandrine has been changed in TTR (lines 99, 184, 344).

  1. Line 115-118: Did the number of MmpL refer in the manuscript are for MmpL only or MmpL and MmpS?

The numbers of MmpL are for MmpL only, as described by Viljoen A et al.: The diverse family of MmpL transporters in mycobacteria: from regulation to antimicrobial developments. Mol Microbiol. 2017; 104: 889-904. (Ref. 35)

  1. Line 127-128: pleas add a reference

The reference 37 has been added in line 136.

  1. Lines 192-198: The author should add the MAB gene number of the efflux pump they are refereeing to. As M. abscessus contained several orthologues of the MmpL5/Mmp5 efflux, this will help the reader to understand these sentences.

The MAB gene number has been added, as requested (line 202).

  1. Lines 236-241: please change homologue for orthologue

As suggested, “homologue” has been changed in “orthologue” in lines 245-248.

  1. Line 302-303: please add a reference

The reference 67 has been added in line 311.

Reviewer 3 Report

the NTM remain an important topic  in today

this organism have found in the many country .

the mycobacterium  may have the initial acid-fast    positive in the NTM and TB.

Therefore, the early detect the NTM and identify the effect drugs for NTM is important.

this draft address the topic about inhibition of the efflux pumps for preventing the resistance of the NTM is an important issue 

the article is novel and interesting .

this draft have merit for publication

Author Response

Responses to Reviewer 3:

I appreciate the general comment by Reviewer.